# Roco Proteins: GTPases with a Baroque Structure and Mechanism

**DOI:** 10.3390/ijms20010147

**Published:** 2019-01-03

**Authors:** Lina Wauters, Wim Versées, Arjan Kortholt

**Affiliations:** 1VIB-VUB Center for Structural Biology, Pleinlaan 2, B-1050 Brussels, Belgium; Lina.Wauters@vub.be; 2Department of Cell Biochemistry, University of Groningen, NL-9747 AG Groningen, The Netherlands; 3Structural Biology Brussels, Vrije Universiteit Brussel, Pleinlaan 2, B-1050 Brussels, Belgium

**Keywords:** Parkinson’s disease, leucine-rich repeat kinase 2, Roco proteins, unconventional G-protein, GTPase mechanism

## Abstract

Mutations in leucine-rich repeat kinase 2 (LRRK2) are a common cause of genetically inherited Parkinson’s Disease (PD). LRRK2 is a large, multi-domain protein belonging to the Roco protein family, a family of GTPases characterized by a central RocCOR (Ras of complex proteins/C-terminal of Roc) domain tandem. Despite the progress in characterizing the GTPase function of Roco proteins, there is still an ongoing debate concerning the working mechanism of Roco proteins in general, and LRRK2 in particular. This review consists of two parts. First, an overview is given of the wide evolutionary range of Roco proteins, leading to a variety of physiological functions. The second part focusses on the GTPase function of the RocCOR domain tandem central to the action of all Roco proteins, and progress in the understanding of its structure and biochemistry is discussed and reviewed. Finally, based on the recent work of our and other labs, a new working hypothesis for the mechanism of Roco proteins is proposed.

## 1. Introduction

Parkinson’s disease (PD) is the second most common neurological disorder after Alzheimer’s disease. 0.3%of the population worldwide suffers from the disease, with an increasing incidence with age [1]. PD is characterized by the progressive loss of dopaminergic neurons in the substantia nigra pars compacta combined with the formation of Lewy bodies (fibrillar protein aggregates rich in α-synuclein) [1,2]. So far, there are no good clinical biomarkers for PD available, and diagnosis is primarily based on clinical symptoms. These symptoms include bradykinesia or slowness of movement, resting tremor, rigidity of the muscles, and postural imbalance [3]. Today PD still cannot be cured or halted, and only symptom improvement can be provided by medication. Almost all available medication focuses on the substitution or increase of dopamine in the brain, with the dopamine precursor levodopa still being the most commonly used PD medication [4,5].

The majority of PD cases are considered to be sporadic or idiopathic [6,7,8]. However, the environmental risk factors for PD are not well known or understood. Nevertheless, several environmental factors, such as pesticide and herbicide exposure, have been positively identified as risk factors for the disease [3,6,9,10]. On the other hand, caffeine consumption was shown to decrease the risk for PD [6,9]. A possible link with a variety of other causative agents like dietary habits, oestrogens, and iron exposure, is still unresolved [3,6]. The less common familial form of PD is caused by autosomal dominant and recessive gene mutations [7]. To date, 23 genes or loci have been associated with familial PD, of which mutations in the gene coding for leucine-rich repeat kinase 2 (LRRK2) are among the most frequent causes [11,12,13]. Moreover, some mutations in LRRK2 have been identified as a risk factor for sporadic PD [7]. Importantly, a recent study by Di Maio et al., showed that LRRK2 kinase activity was enhanced in postmortem brain tissue from patients with idiopathic PD, suggesting that independent of mutations, wild-type LRRK2 plays a role in PD [14]. Therefore, unravelling the working mechanism of LRRK2 can provide significant insights into the pathways leading to neurodegeneration in this disease. LRRK2 is a very large multi-domain protein harboring both GTP hydrolyzing (GTPase) and kinase activity. The protein belongs to a wider family of GTPases, called the Roco protein family, characterized by a central Roc (Ras of complex proteins) and COR (C-terminal of Roc) domain tandem. The Roco protein family was initially described in 2003, and although other family members had been previously investigated, it was only after mutations in LRRK2 were linked to PD that the protein family raised general interest [15].

This review consists of two parts. First, an overview is given of the wide evolutionary range of Roco proteins, leading to their variety in known physiological functions. In the second part, the focus is shifted towards the GTPase function of the central RocCOR domain tandem of Roco proteins. There is an ongoing debate concerning the working mechanism of the RocCOR module. We have already discussed this topic in detail in a previous review [16]. However, here we take into account important recent findings. Based on these new biochemical, biophysical, and structural data, especially on the Roco monomer-dimer equilibrium, we propose a new working hypothesis for the mechanism of Roco proteins in general, and LRRK2 in particular [17,18,19,20]. For more detailed information concerning the structure and function of the kinase and other domains of LRRK2, we refer to other papers and reviews covering these topics [21,22,23,24].

## 2. The Discovery of a Roco Protein Family

The Roco proteins were first discovered in the amoeba *Dictyostelium discoideum* (*D. discoideum*) in 2002 by Goldberg et al., in the form of a cGMP-binding protein (GbpC) with a unique domain architecture including leucine-rich repeats (LRR), a Ras-like domain and a kinase domain [25]. Subsequently, in 2003, Bosgraaf and Van Haastert identified several other proteins with a similar domain architecture. The Ras domains of these proteins clearly differ from other members of the Ras superfamily and were named Roc, after Ras of complex proteins. Furthermore, in all proteins, this Roc domain was followed by a COR, after C terminal of Roc, domain. In this way, the Roco protein family of G-proteins was born, characterized by the presence of a RocCOR domain tandem [26,27].

## 3. A Very Wide Evolutionary Range

After its initial discovery in *D. discoideum* it has been found that the Roco protein family has a very wide evolutionary range, and Roco proteins have been identified in prokaryotes (bacteria and archaea), metazoan, and plants [26,28]. Despite their distinctive RocCOR domain tandem, phylogenetic analysis revealed that the protein family falls apart in three groups based on their domain topology (Figure 1) [26,29,30]. The first group can be found in metazoa, plants, archaea, and bacteria. This group shows the simplest domain arrangement with the RocCOR domain tandem only preceded by an N-terminal LRR domain that is usually involved in protein–protein interactions [26,31,32]. So far all identified prokaryotic Roco proteins belong to this group. Due to their tractability, representatives of this group are often used as model systems for the biochemical and structural study of the functioning of the RocCOR tandem of the Roco protein family [33,34]. Members of the second group, present in metazoa and *D. discoideum*, contain a C-terminal kinase domain in addition to the LRR–RocCOR topology, and often also other regulatory domains [26]. The last group (DAPK1) is found exclusively in metazoa and is characterized by an additional death domain (DD). Remarkably, this is the only group of proteins in which the kinase domain precedes the RocCOR domain, and that does not possess an LRR domain.

## 4. Physiological Functions of Roco Proteins

Very little is known about the physiological functions of Roco proteins in bacteria and archaea. However, the functions of some eukaryotic Roco proteins have been investigated more intensely.

### 4.1. Roco Proteins from Dictyostelium Discoideum

Eleven different Roco proteins have been discovered in *D. discoideum* [26,27]. Using an axenic mutant [35], it was shown that the expression patterns of several Roco genes in this organism are elevated upon starvation, playing a role in the formation of multicellular aggregates [36]. So far, the function of four of these *D. discoideum* Roco proteins has been investigated in more detail. GbpC is a key player in cell aggregation by regulating chemotaxis via myosin II phosphorylation [27,37,38]. Phosphorylated myosin II locates at the rear of polarized cells, where it suppresses the formation of pseudopodia and stimulates the retraction of the rear of the cell [37]. Protein associated with the transduction of signal 1 (Pats1) on the other hand, plays an important role in the recruitment of myosin II to the cleavage furrow of cells during cytokinesis [39,40]. Quick growth protein A (QkgA) has a negative regulatory effect on cell proliferation. *qkgA*-null cells have a faster cell proliferation and slower cell aggregation resulting in the formation of very large slugs [27,36,41]. Finally, Roco4 is another well-studied *D. discoideum* Roco protein. Although wild-type and *roco4*-null cells initially show similar multicellular aggregation upon starvation, *roco4*-null cells form atypical fruiting bodies with a spore head unable to lift into the air due to the absence of cellulose [36]. Together these data show that the Roco proteins in *D. discoideum* have a wide variety of cellular functions despite their similar domain architecture.

### 4.2. Roco Proteins from Vertebrates

Four Roco proteins have been discovered in vertebrates so far, which in humans are called malignant fibrous histiocytoma amplified sequence 1 (MFHAS1), death-associated protein kinase 1 (DAPK1), and leucine-rich repeat kinase 1 (LRRK1) and 2 (LRRK2) [28].

MFHAS1 belongs to the first group of Roco proteins and only consists of an N-terminal LRR domain followed by a central RocCOR domain tandem. On top of the enhanced expression of MFHAS1 in malignant fibrous histiocytomas [42], the gene coding for MFHAS1 was found to be a potential oncogene for gastric, oesophageal, and gastroesophageal cancers [43,44,45]. Despite its oncogenic properties, MFHAS1 is only poorly characterized. A first biochemical study by Dihanich et al. showed that it is a guanosine-5’-triphosphate (GTP)-binding protein with very low endogenous GTPase activity and that the oligomeric state of the protein is influenced by nucleotide binding. However, the precise nature of the observed oligomeric complexes remains to be determined [46].

DAPK1 is part of the third group of Roco proteins, containing a calmodulin-dependent serine/threonine kinase domain, and binding to the cytoskeleton. It is an essential regulator of various cell death signaling pathways and is also important for the activation of autophagy [27,47,48,49]. It has been linked to several neuronal pathologies, including Alzheimer’s disease and ischemia-induced neuronal cell death [50]. DAPK1 also suppresses the cellular transformation during the early stages of tumor development [27,47,51,52]. Deletion or downregulation of DAPK1 causes chronic lymphocytic leukemia [27,53]. The protein forms a dimer through its Roc domain and its calmodulin (CaM)-dependent kinase domain [54,55,56]. Intriguingly, binding of GTP to the Roc domain was reported to have an inhibitory effect on the kinase activity of DAPK1 [54,55]. Upon binding of GTP to the Roc domain, the kinase domain undergoes a conformational change that leads to autophosphorylation of a serine residue in its CaM-regulatory domain. This autophosphorylation decreases the protein’s catalytic activity. Upon GTP hydrolysis, DAPK1 kinase activity is reactivated [50,52,54].

LRRK1 and LRRK2 both belong to the second group of the Roco proteins and have a similar domain topology consisting of N-terminal ankyrin and leucine-rich repeats, followed by a central core of a Roc, COR and kinase domain and finally a C-terminal WD40 domain. LRRK2 has an additional N-terminal armadillo repeat domain preceding the ankyrin repeats [29]. Although LRRK1 and LRRK2 share a similar domain topology, both proteins clearly exhibit different cellular functions [15,27,57,58].

LRRK1 is required for B-cell receptor-mediated B-cell proliferation and survival, and in this way, plays a crucial role in humoral immunity [28,59]. It also has a regulatory function in mitotic spindle orientation. Polo-like kinase 1 phosphorylates LRRK1 on position Ser1790. This activated LRRK1 subsequently phosphorylates the CDK5 regulatory subunit associated protein 2 (CDK5RAP2), which is responsible for centrosome maturation [28,60]. In addition LRRK1 plays a crucial role in epidermal growth factor receptor (EGFR) sorting and transport [28,57,61,62]. The endosomal trafficking of EGFR regulates the amplitude and duration of EGFR signaling [61,62]. Dysregulation of this signaling has been linked to several types of human cancers [62,63]. Activated EGFR is internalized via endocytosis, resulting in the phosphorylation and activation of LRRK1, which in turn regulates the motility of the EGFR-containing endosome. Non-phosphorylated LRRK1 mutants display increased endosome motility and accumulation of EGRF in mixed endosomes [57,62]. On the other hand, LRRK1 also plays a role in the endosomal sorting process by binding EGFR through its adaptor protein growth factor receptor-bound protein 2 (Grb2) and associating it with the ESCRT-0 sorting complex [57,61]. Finally, recent research in LRRK1 knockout mice revealed a negative regulatory role of LRRK1 for bone mass, thus, providing a link between LRRK1 and osteopetrosis [28,59].

Since mutations in the gene coding for LRRK2 have been linked to PD, LRRK2 has become the most intensively investigated Roco protein [15]. LRRK2 localizes both in the cytosol and at specific membrane compartments, where it interacts with a wide variety of proteins. Over the last few years, the protein has been linked to numerous cellular and molecular pathways and functions, including autophagy, cytoskeletal regulation, mitochondrial function, protein translation and degradation, neurite outgrowth and vesicular trafficking. Moreover, accumulating evidence suggests that LRRK2 plays an important role in both the innate and adaptive immunity [64,65].

Both endogenous and exogenously expressed LRRK2 is reported to localize to the mitochondrial outer membrane [66]. Patients with PD-mutated LRRK2 have a decreased mitochondrial membrane potential and reduced cellular ATP levels. Their mitochondria are elongated and have an increased interconnectivity, suggesting an involvement of LRRK2 in mitochondrial fusion and fission [67]. Research by Wang et al. later proved that LRRK2 directly interacts with mitochondrial dynamin-like protein (DLP1) and regulates mitochondrial dynamics using a DLP-mediated pathway [68,69].

LRRK2 was also found to localize to autophagic vacuoles (AVs) and multivesicular bodies (MVBs) [70]. Moreover, expression of PD-mutated LRRK2 in human-derived cell lines leads to an accumulation of AVs and MVBs [70]. This LRRK2-mediated autophagy is suggested to be regulated by the MAPK-ERK signaling pathway and by the calcium-dependent AMPK pathway via LRRK2 activation of NAADP receptors that are involved in the calcium efflux of endosomes [71,72].

A role of LRRK2 in maintaining the neuronal development process has also been reported [73]. Transfection of cortical cultures with PD-mutated LRRK2 with increased kinase activity dramatically reduces neurite length and branching [73,74,75]. Inversely, cells transfected with kinase-deficient LRRK2, show extended structures [73]. LRRK2 regulates neuronal development by, most likely indirectly, stimulating the phosphorylation level of ERM (ezrin, radixin and moesin) proteins, that subsequently regulate axonal growth, cytoskeletal organization and microtubule assembly [76,77,78].

Furthermore, it has been shown that LRRK2 localizes to endosomes where it plays a role in vesicular membrane trafficking [28,79]. Only recently, Steger et al. convincingly identified a subset of Rab GTPases as genuine in vivo LRRK2 substrates [80,81]. Phosphorylation by endogenous LRRK2 has been confirmed for 10 Rab GTPases: Rab3A/B/C/D, Rab5A/B/C, Rab8A/B, Rab10, Rab12, Rab29, Rab35, and Rab43 [80]. Surprisingly, looking at the Rab phylogenetic tree, these Rab proteins are actually widely dispersed. Thus, it might be that the Rabs that are phosphorylated by LRRK2, co-localize with LRRK2 and that this co-localization is the actual phosphorylation determinant [81]. Phosphorylation by LRRK2 influences the cycling of these Rab GTPases between the cytosol and the membrane. Inactive, guanosine-5’-diphosphate (GDP)-bound Rabs are bound by GDP-dissociation inhibitors (GDIs) locking them in the cytosol. Upon prenylation by Rab GGTases, Rab proteins dissociate from their GDI and insert in the membrane, where GDP is exchanged for GTP and the highly conserved threonine residue located in the switch II loop of Rab GTPases is phosphorylated by LRRK2. After GTP hydrolysis and dephosphorylation, the inactive Rabs again bind GDIs and dissociate from the membrane. Overactivation of LRRK2 by PD-mutations result in an increased phosphorylation of Rabs and a reduced affinity for GDIs. This, thus, leads to an accumulation of membrane-bound Rab GTPases which are functionally impaired due to disrupted interaction with effector proteins [80,82,83,84]. It has been reported consistently that LRRK2-mediated phosphorylation negatively regulates Rab signaling. PD-mutated LRRK2 was found to cause a delay in late endosomal trafficking and epidermal growth factor receptor degradation by a decrease in Rab7 activity [79]. Moreover, Maekawa et al. discovered that LRRK2 also has an inhibitory effect on α-synuclein clearance by microglia possibly by decreasing the number of Rab5-positive early endosomes [85]. Finally, it was shown for Rab8a, Rab10, and Rab12 that upon phosphorylation these Rabs bind Rab interacting lysosomal protein like 1 and 2 (RILPL1 and 2), key proteins for ciliogenesis [81]. The phosphorylation of numerous Rab proteins by LRRK2, in combination with their large variety in subcellular localization and the capacity of Rab proteins to insert in subcellular membranes via their lipid group, has led to the emerging theme that LRRK2 recruitment to various membrane structures in cells is controlled by Rab GTPases [64]. It was demonstrated that Rab29 controls a proportion of membrane-bound LRRK2. Moreover, it was shown that Rab29 recruits LRRK2 to the trans-Golgi network and stimulates its kinase activity [83,84]. Another recent study by Eguchi et al. also demonstrated that Rab29 plays a crucial role in recruiting LRRK2 to lysosomes upon stress-induced lysosomal overload. This leads to phosphorylation of Rab8a and Rab10 by LRRK2, causing an attenuation of stress-induced lysosomal enlargement and a promotion of lysosomal secretion, respectively [86]. However, several aspects of these Rab-mediated mechanisms remain to be clarified [64].

Recently, LRRK2 has been closely linked to a function in the immune system. LRRK2 is expressed upon activation of microglia [65,87,88]. Knockdown of LRRK2 results in an attenuated microglial-induced inflammatory cytokine production, while in contrast PD patients show higher levels of activated microglia and an accompanying elevated cytokine response [65,87,88,89]. LRRK2 is also expressed in different cell types of the peripheral immune system and plays a regulatory role in both innate and adaptive immunity [65,90]. An increased expression of LRRK2 was detected in monocytes and macrophages upon stimulation with interferon gamma [65,90,91,92,93]. In dendritic cells, LRRK2 plays a regulatory role in calcium signaling, which is important for the immune function of these cells [65,93,94]. Furthermore, LRRK2 is responsible for the development and maintenance of the B-cell population [65,92,95]. Up to date, two major inflammatory pathways have been linked to LRRK2: the toll-like receptor-mediated and dectin-1 mediated pathways [65,96]. These pathways are responsible for the host immune response against both pathogen-associated and danger-associated self-originating molecules. It is clear that LRRK2 regulates the production of inflammatory cytokines. However, details concerning the exact working mechanism remain to be elucidated [65].

## 5. The Central RocCOR Domain Tandem of Roco Proteins: Structural Insights

Roco proteins are characterized by a central RocCOR domain tandem, responsible for GTP binding and hydrolysis. Sequence alignments with other small GTPases and later crystal structures have shown that the overall architecture of the Roc domain is similar to the classical G-domain. This typical G-domain consists of a central β-sheet surrounded by α-helices and connected by hydrophilic loops. Just as in other G-proteins, five G-domain sequence motifs (G1–5) can be discerned in the Roc domain, which are responsible for nucleotide binding: the phosphate binding loop or P-loop (G1), the switch I motif (G2), the switch II motif (G3), and the guanine specificity determining motifs (G4–G5) [97,98,99,100,101,102,103]. Unlike the first three sequence motifs, the guanine specificity determining motifs (G4–G5) show some variations compared to the classical G-domain [26].

In 2008, the crystal structure was reported of the isolated Roc domain of human LRRK2 in complex with GDP [104]. This structure showed a very unusual dimer, formed through extensive domain swapping (Figure 2). In this domain-swapped dimer, the N-terminal part (G1–3) of one subunit interacts with the C-terminal part (G4–5) of the other and vice versa, forming two hybrid nucleotide binding domains [104].

A later study of the Roc domain by Liao et al., however, claimed this domain swapping to be an artefact of crystallization, since purification of an N- and C-terminally extended Roc domain yielded two peaks in size-exclusion chromatography corresponding to monomeric and dimeric LRRK2 Roc in solution [105]. Moreover, dimerization of the Roc domain was shown to enhance upon phosphorylation [106,107].

A permanent dimerization of the Roc domains via domain swapping was also contradicted by subsequent crystal structures of the COR and RocCOR domain constructs of the Roco protein from the bacterium *Chlorobium tepidum* (*Ct*Roco). Indeed, in 2008 and soon after the publication of the LRRK2 Roc crystal structure, a crystal structure of the RocCOR module of the *C. tepidum* Roco protein in a nucleotide-free state was reported [33]. This structure also shows a dimeric arrangement while only one of the two Roc domains was resolved as an independently folded domain (Figure 3). In contrast, dimerization of the RocCOR in this structure was mainly mediated via interactions between the C-terminal part of the COR domains [33]. Moreover, structural alignment of the domain-swapped human LRRK2 Roc dimer with the bacterial *Ct*RocCOR structure showed that domain swapping would result in severe clashes of the Roc domain with the COR domain, confirming that the Roc domains indeed cannot form a domain-swapped constitutive dimer in the context of the RocCOR [29,105].

Further analysis of the COR and RocCOR structures of *Ct*Roco revealed that the COR domain actually consists of two subdomains connected by a flexible linker (Figure 3). The N-terminal half of the COR region (N-COR) is a mostly α-helical domain with a short three-stranded antiparallel β-sheet. The C-terminal half of COR (C-COR) consists of an antiparallel β-sheet flanked by four α-helices and a β-hairpin motif, the latter forming the main part of the COR dimerization interface [33].

As mentioned before, the crystal structure of *Ct*RocCOR showed two COR domains forming the dimer interface, while only one Roc domain was resolved (Figure 3). The second Roc domain was thought to be present, but presumably, the electron density was very weak due to high mobility. The Roc domain has a typical canonical G-domain fold with an additional N-terminal helix (called α0), which in the context of the full-length protein would connect the Roc domain to the LRR domain. The switch I loop (G2) is flexible and is not visible in the structure. Moreover, sequence alignment of several Roco proteins demonstrated a large variability in length and amino acid composition of this motif. This is in strong contrast with the region around the switch II DXXG motif (G3), which is the most highly conserved region of Roco proteins, together with the residues on the interface between the Roc and COR domain. The switch II loop of Roc is resolved in the structure and makes contact with the most highly conserved patch of the COR domain of the same protomer.

Consistent with the observed implication of the C-terminal half of the COR domain of *Ct*Roco in dimer formation, analytical gel filtration showed that the RocCOR construct elutes at a volume corresponding to a dimer. However, a RocCOR∆C protein construct, which lacks the C-terminal half of the COR domain, elutes at a volume corresponding to a monomer confirming an important role of this part of the protein in dimerization [33]. In 2015, Terheyden et al. reported the crystal structure of a monomeric RocCOR∆C construct, lacking the C-terminal half of the COR domain, from the Roco2 protein from the prokaryotic archaeon *Methanosarcina barkeri* (*Mb*) in the GDP-bound state. Comparison of this structure with nucleotide-free *Ct*RocCOR revealed a rearrangement of the switch II region causing a shift of the N-terminal COR domain. Since it is proposed that G-proteins are molecular switches that cycle between a GTP- and GDP-bound state, and couple this effect to a downstream effector, it might be that the switch II region of the Roc domain plays a role in this intramolecular signaling [34].

Using symmetry arguments, the second Roc domain was modeled in the *Ct*RocCOR structure, resulting in a model where the two Roc domains face each other with their nucleotide-binding sites in a head-to-tail fashion. This observation, thus, suggested that Roco proteins would form constitutive dimers predominantly via their COR domains, while the Roc domains are facing each other. Modeling of the non-hydrolysable GTP-analogue GppNHp into the nucleotide binding site, moreover, showed that the ribose of one nucleotide would face the γ-phosphate of the adjacent nucleotide and vice versa. A further zoom in on the nucleotide binding site revealed the presence of an arginine residue in the Roc domain that could contact the nucleotide binding site of the other Roc domain within the dimer. Mutation of this arginine residue resulted both in *Ct*RocCOR and *Mb*RocCOR in loss of GTPase activity [33,34]. A very recent crystal structure of the LRR–RocCOR domain construct of *Ct*Roco shows that apart from COR–COR interactions the Roc domains also significantly contribute to the dimer interface and reveals a role of the catalytic arginine residue in dimerization [17]. Consistent with these observations in prokaryotes, also in human LRRK2, the RocCOR domain plays an important role in dimerization [34]. However, dimerization was also observed in *full length* LRRK2 constructs that lack the RocCOR domains, suggesting that the RocCOR domain tandem is mainly, but not exclusively, responsible for protein dimerization, most likely the other protein–protein interaction domains can contribute to dimerization as well [56,106].

## 6. Roco Proteins: Conventional G-Proteins, GADs or Yet Another Type of G Proteins?

Based on their working mechanism, G-proteins have been classified either as conventional G-proteins or G-proteins activated by dimerization (GADs) [108]. The catalytic cycle of G-proteins is dependent on both the rate of GTP hydrolysis itself and the rate of nucleotide exchange, i.e., the life-time of the nucleotide-free, GDP- and GTP-bound states. For conventional G-proteins, this cycle is highly regulated by guanine nucleotide exchange factors (GEFs) and GTPase activating proteins (GAPs). These regulators allow G-proteins to switch between an active and inactive state without unnecessary consumption of GTP [97,109,110,111].

Conventional G-proteins typically have very high nucleotide affinities, in the nanomolar to picomolar range. This results in a very slow nucleotide dissociation [97,112,113]. Since in a cell, biological processes occur much faster, within seconds or minutes, than the intrinsic GTP dissociation, these proteins have co-evolved with GEFs. Binding of a GEF to its corresponding G-protein decreases the G-protein’s affinity for the bound nucleotide making nucleotide dissociation possible. GEFs, thus, increase the nucleotide exchange rate (Figure 4a). Another level of regulation is added to the GTP hydrolysis cycle of conventional G-proteins by GAPs. Although conventional G-proteins are GTP hydrolysing enzymes, their actual GTP hydrolysis rate is intrinsically very low. In general, GAPs increase the rate of GTP hydrolysis by orienting a nucleophilic water molecule in an appropriate position for attack on the γ-phosphate of GTP and/or by stabilizing the negative charges of the transition state [109,114].

In contrast to conventional G-proteins, GADs have low micromolar nucleotide affinities, leading to high nucleotide dissociation rates, making GEFs obsolete. Furthermore, when bound to GTP, the G-domains of GADs dimerize and, in this way, complement each other’s active site. Each protomer, thus, provides residues for the active site of the adjacent protomer or has an important stabilizing role on its active site. These residues can stabilize the flexible active site or can be directly involved in catalysis. Either way, GADs have all the necessary components for GTP hydrolysis and are not strictly dependent on GAPs for GTP hydrolysis (Figure 4b) [108].

The exact GTPase mechanism of Roco proteins still forms the subject of debate due to the lack of structural information on the active GTP-bound conformation. In support of the conventional G-protein theory, some possible GEFs and GAPs have been reported for LRRK2 [115]. Rho guanine nucleotide exchange factor ARHGEF7 was shown to interact and partially co-localize with LRRK2 in vitro and in vivo. It enhances the exchange of GDP for GTP and in this way, might function as a GEF for the GTPase activity of LRRK2. It should, however, be noted that ARHGEF7 does not bind directly to the Roc domain of LRRK2, unlike conventional GEFs. Strikingly, ARHGEF7 is also reported to be an in vitro substrate of LRRK2, with two threonine residues at the N-terminus of ARHGEF7 being phosphorylated by LRRK2 [116,117].

Two potential LRRK2 GAPs were identified as well: ArfGAP1 and RGS2 [118,119]. The ArfGAP1-LRRK2 interaction has been demonstrated both in vitro and in vivo in brain tissue [118]. Binding of ArfGAP1 has no effect on LRRK2’s GTP binding capacity but mediates LRRK2 toxicity by increasing the GTP hydrolysis rate. However, ArfGAP1 binds LRRK2 primarily via the WD40 and kinase domain, rather than via the Roc GTPase domain [120]. In accordance, LRRK2 has been shown to phosphorylate ArfGAP1. Studies concerning the effect of ArfGAP1 on LRRK2 kinase activity and the other way around, have however given controversial results. In different conflicting studies, ArfGAP1 has been shown to either reduce or induce LRRK2 kinase activity. In addition, the role of phosphorylation of ArfGAP1 is unclear [118,120]. In 2014, Dusonchet et al. identified a second possible LRRK2 GAP, named RGS2. RGS2 increases the GTPase activity of LRRK2 in vitro. Surprisingly, independent of its influence on the GTPase activity, RGS2 was shown to have an inhibitory effect on LRRK2 kinase activity and in this way controls neurite length. RGS2 is also a substrate for phosphorylation by LRRK2 in vitro. Future studies are needed to investigate whether RGS2 is also a substrate of LRRK2 in vivo and whether RGS2 is a physiological GAP for LRRK2 in vivo, or mediates its function via another mechanism [117,121]. In fact, for all GEFs and GAPs reported, their direct interaction with the LRRK2 Roc domain is still not proven. Hence, ARHGEF7, RGS2, and ArfGAP1 could modulate the GTPase activity of LRRK2, but most likely function in a manner distinct from classical GEFs and GAPs [115].

On the other hand, based on the above-discussed structures and the dimeric nature of Roco family proteins, including LRRK2, it was postulated that Roco proteins belong to the GAD family of proteins [33,34,108,122]. Moreover, we recently showed that full length prokaryotic Roco proteins have a low, micromolar affinity for GDP (9–55 µM) associated with fast GDP dissociation rates (12.8–230 min^−1^) [18]. This is in line with the reported affinity of the human LRRK2 Roc domain for GTP and GDP of, respectively, 7.85 and 0.47 µM [105]. Based on these studies it can be concluded that GEFs are not strictly necessary for the functioning of Roco proteins. The in vitro GTP hydrolysis rate of Roco proteins (0.06–0.5 min^−1^) [18] lies well within the range of the basal activity of GADs, like MnmE (0.33 min^−1^), as well as within the intrinsic GTP hydrolysis activity of conventional G-proteins, such as Ras (0.03 min^−1^) [123,124]. Whereas the basal activity of Ras is stimulated up to 300 to 600 min^−1^ by RasGAP, the GTP hydrolysis rate of MnmE increases up to 9.3 min^−1^ upon interaction with K^+^ ions and its interaction partner MnmG [125]. It is, thus, possible that the moderate GTPase activity of Roco proteins is indeed triggered by interaction with cofactors, including RGS2 and ARFGAP1 [119,121,126].

The kinetic data, combined with the dimeric prokaryotic RocCOR protein crystal structures, thus, argue against a model where Roco proteins function as conventional G-proteins. Moreover, a wide variety of assays such as tandem affinity purifications, yeast two-hybrid assays, pull downs, co-immunoprecipitation, size-exclusion chromatography, single-particle transmission electron microscopy and immunogold labelling using cell extracts or purified LRRK2 confirmed a dimeric nature of LRRK2 under many circumstances [15,56,127,128]. Gel filtration and blue native gels using cell lysates suggested that the majority of LRRK2 was dimeric, with a small fraction being monomeric or forming higher oligomers [82,129]. Furthermore, it was shown that the GTPase activity of the full length LRRK2 (k_cat_ = 0.5 min^−1^; K_M_ = 400 µM) or its RocCOR–kinase construct (k_cat_ = 0.8 min^−1^; K_M_ = 343 µM) is 25 to 40 times higher compared to Roc alone (k_cat_ = 0.02 min^−1^; K_M_ = 553 µM), suggesting that dimerization of LRRK2 indeed might increase its GTPase activity, although other factors can also contribute to the observed low activity of the isolated Roc domain [18,105,130].

Despite these pieces of evidence that purified Roco proteins behave mainly as dimers, cell fractionation experiments using endogenous or exogenous expressed LRRK2 revealed that the majority of LRRK2 in cells is cytosolic and monomeric, while only a small portion of dimeric LRRK2 is localized at the membrane. The membrane-associated dimeric LRRK2 shows a higher kinase activity, increased GTP-binding capacity and a decrease in phosphorylation [82,129]. Moreover, the first study in living cells, using confocal and total internal reflection microscopy coupled to the number and brightness analysis, confirmed the monomeric nature of LRRK2 in the cytosol and the formation of higher oligomers in the plasma membrane [106,131]. Recent work by our research groups further investigated the oligomeric state of Roco proteins during GTP hydrolysis in vitro. By combining small-angle X-ray scattering, size-exclusion chromatography coupled to multi-angle light scattering, sedimentation velocity analytical ultracentrifugation, native mass spectrometry and electron microscopy it was shown that *Ct*Roco is mainly monomeric when bound to GTP, dimeric in its nucleotide-free state and that an intermediate state can be observed upon binding to GDP. Our very recently published structure of a LRR–RocCOR domain construct of *Ct*Roco, in combination with hydrogen-deuterium exchange coupled to mass spectrometry analysis, revealed that these nucleotide-dependent changes in oligomerisation are relayed to conformational changes among the LRR, Roc, and COR domains via the Roc switch II [17]. The different oligomeric states are also observed during a round of GTP hydrolysis and form an integral part of the GTP hydrolysis cycle [19]. In parallel, a kinetic characterization of full length *Ct*Roco has demonstrated that neither nucleotide dissociation nor P_i_ release are rate-limiting, but that GTP hydrolysis itself, or associated conformational changes after nucleotide binding but prior to product release, form the rate-limiting step in the GTP hydrolysis cycle [18].

## 7. Proposal of a New Working Mechanism for Roco Proteins

Based on the above discussed in vitro and in vivo results, we can hypothesize on a new working model for the mechanism of Roco proteins in general, and LRRK2 in particular (Figure 5). GTP-bound monomeric LRRK2 is uniformly distributed in the cytosol. A state that is stabilized by 14-3-3 proteins which bind to LRRK2 upon phosphorylation of its N-terminally located serine residues and prevent LRRK2 aggregation in cytosolic inclusion pools [132,133,134]. In this stabilized monomeric state, both the GTPase and kinase presumably have only low basal activity [82].

In addition, inactive GDP-bound Rab proteins are maintained in the cytosol by binding to GDP dissociation inhibitors (GDIs). Interaction with a Rab-escort protein (REP), facilitates prenylation of these Rab proteins by Rab GGTases, which results in detachment of GDIs and membrane-binding [80,82,83]. Here, the aid of GEFs results in a rapid exchange of GDP for GTP, resulting in GTP-bound Rab proteins at the membrane in their active state. These GTP-bound Rab proteins, recruit GTP-bound LRRK2 to the membrane, by binding to the N-terminus of LRRK2. This recruitment has for example been demonstrated for Rab29, which binds the Ankyrin domain of LRRK2 and in this way, recruits the protein to the Golgi apparatus. Upon membrane-binding, the LRRK2 kinase is activated by so far not completely understood mechanisms and phosphorylates prenylated, GTP-bound Rab proteins, as well as its own serine 1292 residue. This Rab phosphorylation hinders interaction with RabGAPs, and as such, slows down GTP hydrolysis of Rab proteins [83,84].

Meanwhile or prior to the activation of the kinase domain, membrane-association of LRRK2 also induces LRRK2 GTP hydrolysis, which could, for example, be triggered by the release of 14-3-3 proteins, conformational changes caused by lipid binding, a local higher protein concentration or by binding of a membrane-bound GAP. This step is followed by protein dimerization, resulting in a dimeric, GDP-bound conformation of LRRK2 at the membrane. So far it remains to be determined if GTP binding itself is sufficient to trigger monomerization and detachment from the cell membrane or that additional co-factors are necessary to regulate this process [19].

Several studies have reported that PD mutations in the Roc and COR domain result in decreased GTPase activity. However the underlying defects in the activation mechanism were not well understood [12,56,104,105,129,135,136,137,138,139]. Our recent work with the *Ct*Roco protein provides a link between PD-mutations, decreased GTPase activity and changes in the monomer/dimer equilibrium [19]. A recent study by Purlyte et al. links this decreased GTPase activity in R1441G/C and Y1699C mutant LRRK2 to an increased kinase activity, since mutant LRRK2 stays longer in its kinase active, membrane-bound conformation [84].

## 8. Perspectives

The above-described model clearly generates new insights, but also raises many new important questions. Although it has been shown that *Ct*Roco switches between a monomeric and dimeric state during GTP hydrolysis, the structural mechanism and functional implications for this transition remains to be determined [19]. Interestingly, in contrast to many other small Ras-like GTPases which display very high affinity for GTP and GDP, our recent data suggest that the GTPase activity of Roco proteins might be directly dependent on the physiological levels of GTP [18]. The conserved metabolite GTP drives chemical reactions in the cell. However, the cellular status strongly influences the concentration of GTP, which varies between 0.1 and 1 mM [123,124]. The Michaelis–Menten constant (K_M_) of the GTPase reaction of full length Roco proteins and LRRK2 lies in a similar range as the cellular concentration of GTP, suggesting that the rate of GTP hydrolysis will scale with the GTP concentration and that Roco proteins might, thus, act as GTP sensors rather than as classical switches. However, future research is clearly needed to further explore this potential function of Roco proteins [100,126]. Moreover, it is as yet unclear whether GTP hydrolysis and protein re-dimerization coincide or are actually coupled to each other. Furthermore, although PD analogous mutations have been shown to affect the Roco monomer/dimer equilibrium, further research is necessary to confirm that a similar mechanism actually holds true for LRRK2. The PD mutations within the LRRK2 Roc (R1441C/G/H) and COR (Y1699C) all have reduced GTPase activity. However, the exact effect of these PD mutations on dimerization remains to be determined. Interestingly, modelling of the LRRK2 pathogenic mutations on our recently *Ct*LRR–Roc–COR structure revealed that these mutations are located within the same regions that are important for nucleotide-dependent monomer-dimerization [17]. According to our model, Roco proteins and presumably LRRK2 need to cycle through a monomeric and dimeric state, thus, stabilization by PD-mutations of either state could lead to the observed reduced GTPase activity of the PD mutants. The link between the imbalance in homodimerization and PD, thus, creates an appealing drug target [19]. Besides the RocCOR domain tandem, LRRK2 also contains a kinase domain and several protein–protein interaction domains [133]. Both the GTPase and the kinase function of LRRK2 are essential, and depend on each other, for proper functioning [12,128]. Further research is needed to unravel how these two domains interact and how additional regulatory factors like phosphorylation, binding to effectors and cellular localization influence the protein’s functioning and oligomerization [107,136,140,141].

## Figures and Tables

**Figure 1 ijms-20-00147-f001:**
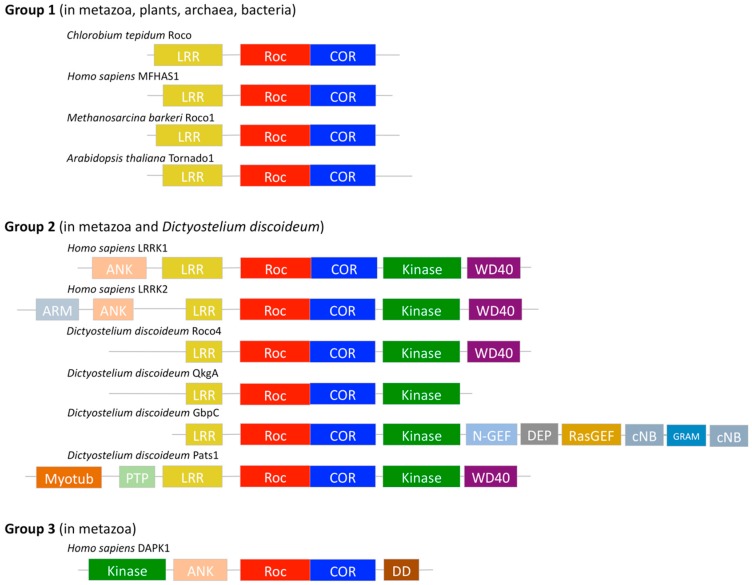
Classification of Roco proteins based on domain topology. Roco proteins are characterized by the presence of a Roc domain (red) and COR domain (blue). Based on their domain topology, Roco proteins can be divided into three groups [29]. Here some representatives of each group are shown. The first group only contains an LRR domain (yellow) preceding the RocCOR. Representatives of this group are found in metazoa, plants, archaea, and bacteria. The second group contains, in addition to this LRR–Roc–COR domain topology, a C-terminal kinase domain (green). Moreover, members of this group contain several protein–protein interaction and regulatory domains like an Ankyrin repeat (ANK, pink), armadillo repeats (ARM, light blue), a N-terminal motif of RasGEF (N-GEF, light blue), a cyclic nucleotide binding domain (cNB, light grey), a Rab-like GTPase activators and myotubularins domain (GRAM, blue), a Ras Guanine Exchange Factor domain (RasGEF, beige), a N-terminal myotubularin-related domain (myotub, orange), a protein tyrosine phosphatase domain (PTP, light green), a Dishevelled domain, and an Egl-10 domain and Pleckstrin domain (DEP, grey). The last group has one human family member, DAPK1. DAPK1 lacks an LRR domain and is characterized by its C-terminal death domain (DD, brown).

**Figure 2 ijms-20-00147-f002:**
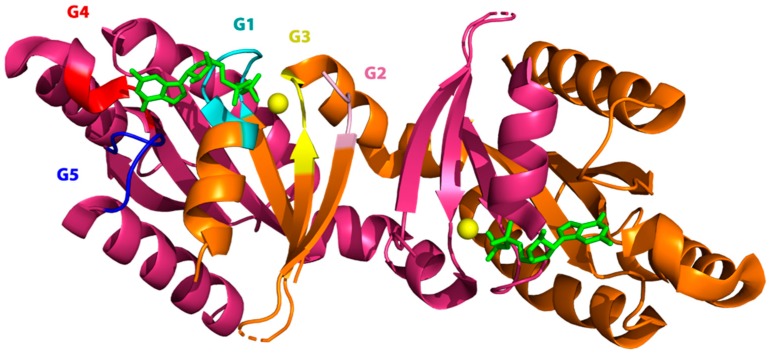
Crystal structure of the swapped dimer of the LRRK2 Roc domain bound to GDP (pdb: 2zej) [104]. The two swapped G-domains are shown in orange and pink, respectively. For one active site the P-loop or G1 motif (cyan), the switch I or G2 motif (light pink), the switch II or G3 motif (yellow), the G4 (red), and G5 motif (blue) are highlighted. Each active site is composed of the G1–3 motif of one protomer and the G4–5 motif of the other. GDP is depicted as a green stick model an Mg^2+^ as a yellow sphere.

**Figure 3 ijms-20-00147-f003:**
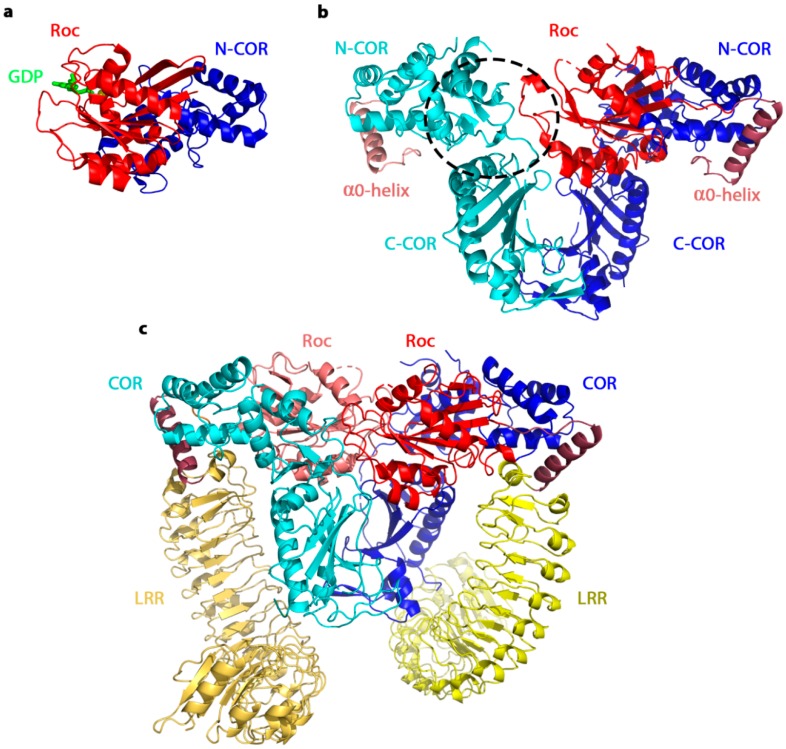
The C-terminal half of the COR domain is important for dimerization (pdb: 3DPU, 4WNR and 6HLU). (**a**) The crystal structure of the RocCOR∆C domain construct from the Roco2 protein of *Methanosarcina barkeri* reveals a monomeric conformation [33]. (**b**) The crystal structure of the RocCOR domain construct from the Roco protein of *Chlorobium tepidum* shows dimer formation via the C-terminal COR domains. The COR domain contains two subdomains: an N-terminal, mainly α-helical domain with a short antiparallel β-sheet (N-COR), and a C-terminal domain with a β-sheet surrounded by α-helices and a β-hairpin involved in dimerization (C-COR). Only one Roc domain is resolved in the structure, presumably due to the flexibility of the other Roc domain. The putative site of the second Roc domain is indicated with a black dotted line [33]. (**c**) The crystal structure of the LRR–RocCOR domain construct from *Ct*Roco. This structure reveals the second Roc domain and the orientation of the LRR domains with respect to the other domain. Apart from the COR–COR interactions, also Roc–Roc and Roc–COR interactions contribute significantly to the dimer interface [17].

**Figure 4 ijms-20-00147-f004:**
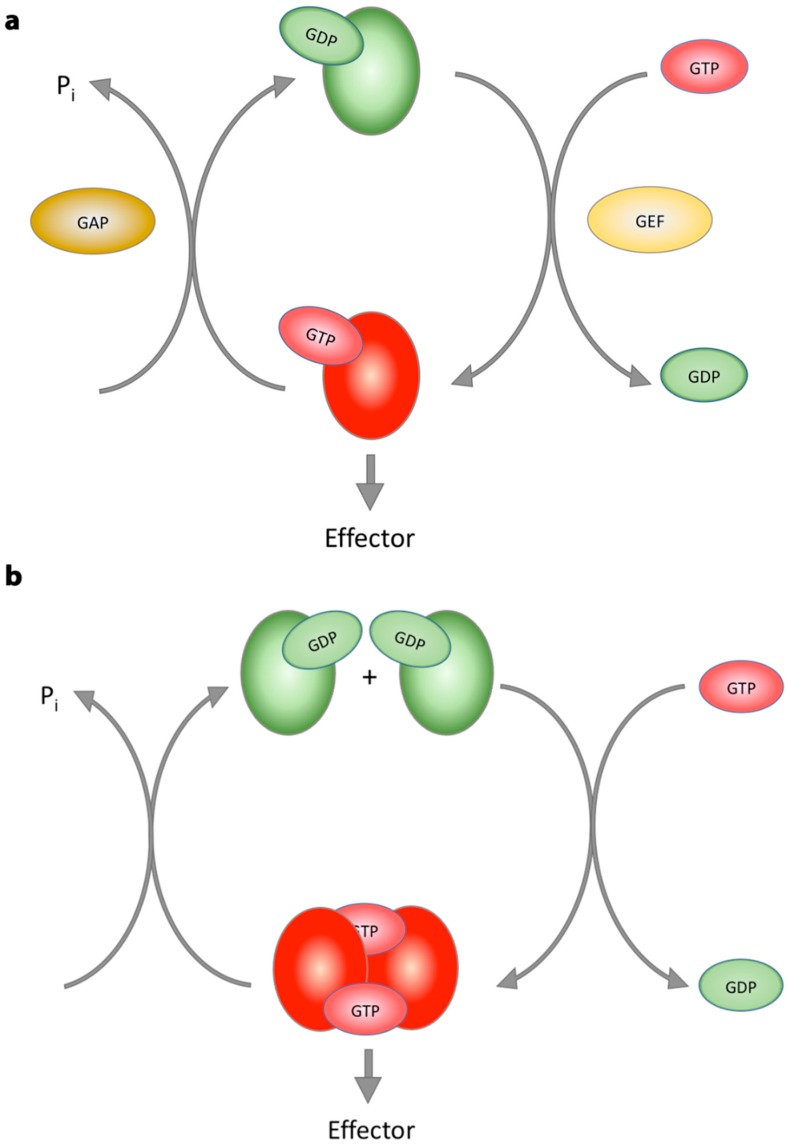
G-protein cycle of conventional G-proteins versus G-proteins activated by dimerization (GADs). (**a**) Conventional G-proteins have very high nucleotide affinities resulting in a low nucleotide dissociation rate. Binding of guanine nucleotide exchange factors (GEFs) decreases the affinity for the bound nucleotide allowing it to be released from the protein. Due to the higher cellular GTP concentration, GTP then binds to the protein and the protein switches to its active GTP-bound state, where it interacts with downstream effectors. GTPase activating proteins (GAPs) can then bind to the G-protein, stabilize and/or complement its catalytic machinery and in this way, increase its intrinsically low GTP hydrolysis rate. The G-protein then switches to its GDP-bound inactive state. (**b**) GADs have micromolar nucleotide affinities, leading to high nucleotide dissociation rates. Therefore, they do not require GEFs to cycle from their inactive GDP- to their active GTP-bound state. Following GTP binding, the G-domains of GADs dimerize. In this way, both subunits complement each other’s active site and are able to hydrolyse GTP. The GADs then cycle back to their inactive, GDP-bound state. In conclusion, GADs possess all the necessary components for GTP-binding and hydrolysis and cycle between an active and inactive state without the aid of GEFs and GAPs [108].

**Figure 5 ijms-20-00147-f005:**
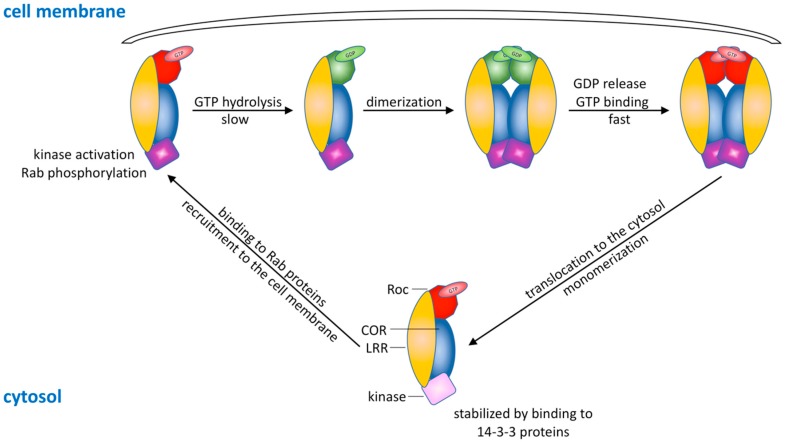
A newly proposed working mechanism for LRRK2. Cytosolic GTP-bound monomeric LRRK2 is recruited to the cell membrane by binding via its N-terminal domains to GTP-bound Rab proteins that are located at the membrane. At the cell membrane, GTP is hydrolysed, and the protein dimerizes. Meanwhile, the LRRK2 kinase domain is activated, and Rab proteins are phosphorylated. The low affinity of LRRK2 for GDP then leads to fast GDP release. Due to the higher GTP concentration present in the cell, this results in rebinding of GTP, monomerization of LRRK2 and return to the cytosol.

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
