# Peer review of "Roco Proteins: GTPases with a Baroque Structure and Mechanism"

_ijms, 2019, doi:10.3390/ijms20010147_

Round 1

Reviewer 1 Report

The review by Wauters et al.describes the role of Roco proteins with a focus on the PD protein LRRK2 even-handedly. The authors are well known experts in the field and cover all aspects of the topic in a comprehensive manner while providing their own aspect to guide readers along. I genuinely enjoyed reading the review and have very little to add and only found a couple of minor corrections. The presented figures are very helpful in providing an excellent overview on the structure and function of Roco proteins and LRRK2. 

Minor correction required:

1. Please correct Alzheimer’s Disease to Alzheimer’s disease (line 24 and 134)

2. Please reconsider the statement ‘So far no diagnostic test is available for the detection of PD, and . . .’ (line 27) I assume the statement is meant to indicate the lack of easily to the clinic translatable biomarkers for PD such as the ones detectable by a ‘blood test’. However, it is correct that PD is mainly diagnosed clinically e.g. by testing the response of patients to L-DOPA. In addition, PET imaging becomes a good indicator for PD. However, admittedly this technique is not commonly used to diagnose PD.

3. Please reconsider the statement ‘… levodopa, a dopamine precursor, being the most effective’ (line 31). This is more complicated dependent on the desired outcome and clinical presentation of the patient. I would suggest stating that levodopa is still the most commonly used PD medication.

4. Please reconsider the statement that tobacco decreases the risk of PD. In fact, the environmental risk factors for PD are not well known or understood. However, in particular concerning the interaction of tobacco and PD, publication bias should be considered, not to mention the fact that smokers might have a decreased chance of getting PD as they are less likely to reach the age of PD onset due to smoking related diseases. In addition, reverse association might also be a reason for the observed association i.e. people developing PD might be more risk averse.

5. Line 65/66 please correct: ‘…in all proteins this Roc domain was proceeded by a COR . . .’ to ’ ‘…in all proteins this Roc domain was followed by a COR . . .’

6. Line158 please correct: ‘…, which in itsturn ... ‘ to ‘‘…, which in turn . ..’.

7. Please reconsider the statement ‘LRRK2 regulates neuronal development by phosphorylating ERM proteins …’ Even though the citation is correct the cited authors (72) later acknowledged that moesin, ezrin and radixin are unlikely to be direct LRRK2 phosphorylation substrates. This should be mentioned. This does not exclude any indirect effect of LRRK2 on the phosphorylation of these proteins.

8. Line 216 please correct ‘Rab29 controls the proportion . . .’ to ‘Rab29 controls proportion . . .’

Author Response

Reviewer #1: The review by Wauters et al.describes the role of Roco proteins with a focus on the PD protein LRRK2 even-handedly. The authors are well known experts in the field and cover all aspects of the topic in a comprehensive manner while providing their own aspect to guide readers along. I genuinely enjoyed reading the review and have very little to add and only found a couple of minor corrections. The presented figures are very helpful in providing an excellent overview on the structure and function of Roco proteins and LRRK2.

Minor correction required:

1.     Please correct Alzheimer’s Disease to Alzheimer’s disease (line 24 and 134)

We have corrected this in the revised manuscript

2.     Please reconsider the statement ‘So far no diagnostic test is available for the detection of PD, and . . .’ (line 27) I assume the statement is meant to indicate the lack of easily to the clinic translatable biomarkers for PD such as the ones detectable by a ‘blood test’. However, it is correct that PD is mainly diagnosed clinically e.g. by testing the response of patients to L-DOPA. In addition, PET imaging becomes a good indicator for PD. However, admittedly this technique is not commonly used to diagnose PD.

We agree with the reviewer and have changed this to: “So far, there are no good clinical biomarkers for PD available, and diagnosis is primarily based on clinical symptoms”

3.     Please reconsider the statement ‘… levodopa, a dopamine precursor, being the most effective’ (line 31). This is more complicated dependent on the desired outcome and clinical presentation of the patient. I would suggest stating that levodopa is still the most commonly used PD medication.

We agree with the reviewer and have changed this accordingly.

4.     Please reconsider the statement that tobacco decreases the risk of PD. In fact, the environmental risk factors for PD are not well known or understood. However, in particular concerning the interaction of tobacco and PD, publication bias should be considered, not to mention the fact that smokers might have a decreased chance of getting PD as they are less likely to reach the age of PD onset due to smoking related diseases. In addition, reverse association might also be a reason for the observed association i.e. people developing PD might be more risk averse.

We agree with the reviewer and have rewritten this paragraph and removed the claim about smoking.

5.     Line 65/66 please correct: ‘…in all proteins this Roc domain was proceeded by a COR . . .’ to ’ ‘…in all proteins this Roc domain was followed by a COR . . .’

We have corrected this in the revised manuscript

6.     Line158 please correct: ‘…, which in itsturn ... ‘ to ‘‘…, which in turn . ..’.

We have corrected this in the revised manuscript.

7.     Please reconsider the statement ‘LRRK2 regulates neuronal development by phosphorylating ERM proteins …’ Even though the citation is correct the cited authors (72) later acknowledged that moesin, ezrin and radixin are unlikely to be direct LRRK2 phosphorylation substrates. This should be mentioned. This does not exclude any indirect effect of LRRK2 on the phosphorylation of these proteins.

We thank the reviewer for pointing this out. In the revised manuscript we changed this to: “LRRK2 regulates neuronal development by, most likely indirectly, stimulating the phosphorylation level of ERM (ezrin, radixin and moesin) proteins, that subsequently regulate axonal growth, cytoskeletal organization and microtubule assembly [72–74].”

8.     Line 216 please correct ‘Rab29 controls the proportion . . .’ to ‘Rab29 controls proportion . . .’

We have corrected this in the revised manuscript.

Reviewer 2 Report

This is an interesting summary of Roco GTPases and LRRK functions. It has thoughtful discussion of current understanding and perspectives. 

Phylogenic trees could be constructed across different Roco and LRRK proteins so that the readers can infer their relationships.

This is a thorough biochemical review. It could be also interesting if the authors can briefly summarize a pathogenetic mechanism (or at least possibilitie) of how LRRK defects can cause PD and how the new model of GTPase regulation can shed a light into that understanding.

The authors have published a review paper with a very similar title (PMID: 27913669). Although the current one is more extensive with some additional insights (with discussion of more recent findings), the authors should make it clear that they have discussed about this topic previously (with citation) and that how this new review is different from the former one and will update the information to the field additionally. For instance, what are the most important updates in the last 2-3 years? These should be mentioned in the introduction. In addition, the reviewer thinks that the authors should change their title to focus on these major updates?

Author Response

Reviewer #2: This is an interesting summary of Roco GTPases and LRRK functions. It has thoughtful discussion of current understanding and perspectives.

Specific comments:

1.     Phylogenic trees could be constructed across different Roco and LRRK proteins so that the readers can infer their relationships.

Phylogenic trees have been published for Roco and LRRK2 and therefore it is in our opinion not necessary to include this in the current manuscript. On page 3 we now refer to several of these phylogenetic studies.

2.     This is a thorough biochemical review. It could be also interesting if the authors can briefly summarize a pathogenetic mechanism (or at least possibilitie) of how LRRK defects can cause PD and how the new model of GTPase regulation can shed a light into that understanding

We extended our discussion on this topic (see perspectives on page 21), however since clearly more research is necessary to completely understand the pathogenic mechanism we don’t want to speculate too much.

3.     The authors have published a review paper with a very similar title (PMID: 27913669). Although the current one is more extensive with some additional insights (with discussion of more recent findings), the authors should make it clear that they have discussed about this topic previously (with citation) and that how this new review is different from the former one and will update the information to the field additionally. For instance, what are the most important updates in the last 2-3 years? These should be mentioned in the introduction. In addition, the reviewer thinks that the authors should change their title to focus on these major updates?

We have added this to the introduction according the reviewers suggestions and changed the title to: “Roco proteins: GTPases with a baroque structure and mechanism”.

Round 2

Reviewer 2 Report

The responses are acceptable.